**Article** https://doi.org/10.1038/s41467-022-34043-9

# Observation of gapped Dirac cones in a two-dimensional Su-Schrieffer-Heeger lattice

Daiyu Geng [1,2,5], Hui Zhou[1,2,5], Shaosheng Yue [1,2], Zhenyu Sun[1,2], Peng Cheng [1,2], Lan Chen [1,2,3], Sheng Meng [1,2,3,4] ✉, Kehui Wu [1,2,3,4] ✉ & Baojie Feng [1,2,4] ✉

The Su-Schrieffer-Heeger (SSH) model in a two-dimensional rectangular lattice features gapless or gapped Dirac cones with topological edge states along specific peripheries. While such a simple model has been recently realized in photonic/acoustic lattices and electric circuits, its material realization in condensed matter systems is still lacking. Here, we study the atomic and electronic structure of a rectangular Si lattice on Ag(001) by angle-resolved photoemission spectroscopy and theoretical calculations. We demonstrate that the Si lattice hosts gapped Dirac cones at the Brillouin zone corners. Our tight-binding analysis reveals that the Dirac bands can be described by a 2D SSH model with anisotropic polarizations. The gap of the Dirac cone is driven by alternative hopping amplitudes in one direction and staggered potential energies in the other one and hosts topological edge states. Our results establish an ideal platform to explore the rich physical properties of the 2D SSH model.

The discovery of graphene has inspired enormous research interest in searching for topological materials with Dirac cones[1–5]. In a Dirac material, the valence and conduction bands touch at discrete points in the momentum space, and the band degeneracies are protected by symmetries, such as time-reversal and mirror reflection. The topological band structures of Dirac materials can give rise to various exotic properties, including the half-integer quantum Hall effect[6], Klein tunneling[7], and extremely large magnetoresistance[8]. Compared to three-dimensional Dirac materials, two-dimensional (2D) Dirac materials are relatively rare because of the scarcity of realizable 2D materials[9]. In addition, most of the experimentally realized 2D Dirac materials are hexagonally symmetric, such as graphene[10,11] and silicene[12,13]. In rectangular or square lattices, Dirac states are quite rare[14–16], despite the recent prediction of candidate materials including 6,6,12-graphyne[17,18], $t_1(t_2)$-SiC[19], and g-SiC$_3$[20,21]. On the other hand, breaking certain symmetries can gap out the Dirac point, giving rise to topological (crystalline) insulating states with conducting edge channels.

A prototypical model to realize topological states is the Su-Schrieffer-Heeger (SSH) model, which was initially put forward to describe spinless electrons hopping in a one-dimensional (1D) dimerized lattice[22]. Recently, the 1D SSH model has been extended to 2D square lattices[23–25]. In the 2D SSH model, a gapless Dirac cone exists at the M point when electron hoppings are homogeneous, as shown in Fig. 1a, b. Suppose the hopping amplitudes alternate in one direction while keeping constant in the other one. In that case, the application of a staggered potential in the other direction can gap out the Dirac cone, as shown in Fig. 1c, d, with topological edge states along specific peripheries. Experimentally, the 2D SSH model has been realized in several systems, including photonic/acoustic lattices[26–28] and electric circuits[29]. However, in condensed matter systems, the material realization of 2D SSH lattice is still lacking.

In this work, we demonstrate that Dirac electronic states can be realized in a rectangular Si lattice, and the topological properties of this system can be described by the 2D SSH model. The rectangular Si lattice can be synthesized by epitaxial growth of Si on Ag(001), which

[1]Institute of Physics, Chinese Academy of Sciences, 100190 Beijing, China. [2]School of Physical Sciences, University of Chinese Academy of Sciences, 100049 Beijing, China. [3]Songshan Lake Materials Laboratory, 523808 Dongguan, Guangdong, China. [4]Interdisciplinary Institute of Light-Element Quantum Materials and Research Center for Light-Element Advanced Materials, Peking University, 100871 Beijing, China. [5]These authors contributed equally: Daiyu Geng, Hui Zhou. ✉e-mail: smeng@iphy.ac.cn; khwu@iphy.ac.cn; bjfeng@iphy.ac.cn

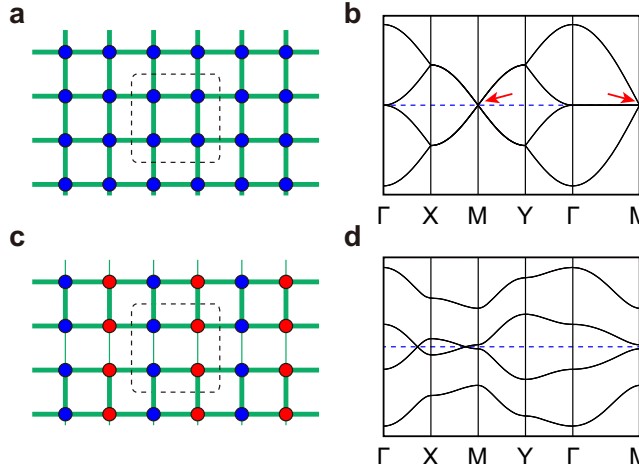

**Fig. 1 | Illustration of the 2D SSH model. a, c** Schematic drawing of the 2D SSH model. The rectangle lattice has homogeneous hopping (**a**) and alternative hopping in *y* direction and staggered potential energy in *x* direction (**c**), respectively. The hopping strengths of electrons are illustrated by the thickness of the bonds. The color of different atoms indicates different on-site energy. **b, d** Calculated band structures of (**a**) and (**c**), respectively. Red arrows in **b** indicate the gapless Dirac cones at the M point.

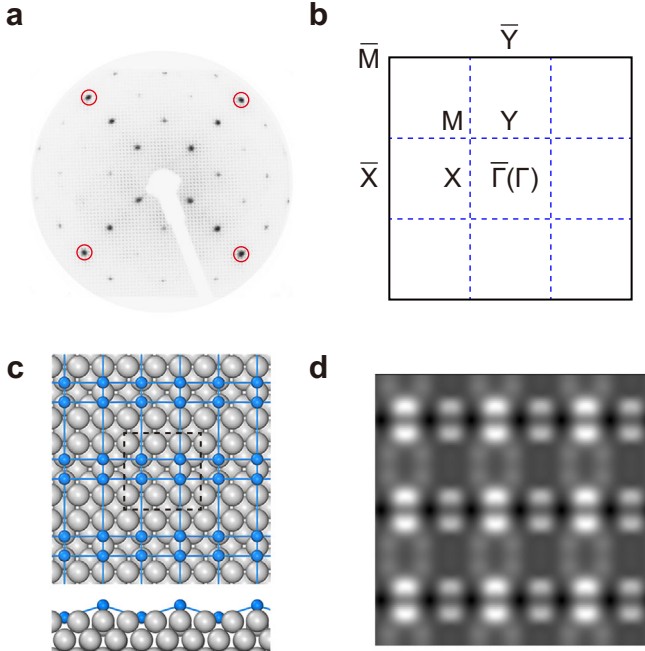

**Fig. 2 | Structure of the Si/Ag(001) system. a** LEED pattern of Si grown on Ag(001). Red circles indicate the diffraction patterns from the 1 × 1 lattice of Ag(001). Si forms a 3 × 3 superstructure with respect to the substrate. **b** Schematic drawing of the BZs of Si and Ag(001). **c** Relaxed structure model of Si/Ag(001). Blue and gray balls indicate the Si and Ag atoms, respectively. **d** Simulated STM image of Si/Ag(001) at bias voltage of −1 eV.

was reported as early as 2007[30]. Previous works on this system only focused on the structural characterization using scanning tunneling microscopy (STM) and surface X-ray diffraction, while studies of its electronic structure are still lacking, both experimentally and theoretically.

Here, we systematically study the atomic and electronic structure of the rectangular Si lattice by ARPES measurements, first-principles calculations, and tight-binding analysis. Our first-principles calculations show that adsorption of Si on Ag(001) will result in the periodic missing of topmost Ag atoms, in contrast to the previously proposed unreconstructed Ag(001) surface[30,31]. Interestingly, our ARPES measurements proved the existence of a gapped Dirac cone at each M point of the Si lattice, which is supported by our first-principles calculations. Our tight-binding (TB) analysis based on a 2D SSH model reveals that the gap of the Dirac cone is driven by the alternation of bond lengths in one direction and stagger of potential energies in the other one. In addition, our calculations show that topological edge states exist along specific peripheries, which indicates rich topological properties in this system. These results call for further research on the exotic topological properties of the 2D SSH model in condensed matter physics.

## Results and discussion

### Growth and atomic structure

When the coverage of Si is less than one monolayer, a 3 × 3 superstructure with respect to the Ag(001) substrate will form[30]. The sharp LEED patterns shown in Fig. 2a indicate the high quality of the sample. Further deposition of Si will lead to the formation of the second layer, which is a more complex phase compared to the first layer[30]. Here, we focus on the monolayer phase, i.e., the 3 × 3 superstructure. Figure 2b schematically shows the Brillouin zones (BZs) of the Si lattice and the Ag(001) substrates together.

To determine the atomic structure of the Si/Ag(001) system, we performed first-principles calculations. Previously, Leandri et al.[30] suggested that the 3 × 3 superstructure is formed by the periodic arrangement of tilted Si dimers on an unreconstructed Ag(001) surface. However, this structure model is neither stable nor metastable according to first-principles calculations[31]. He

et al. proposed that Si atoms form hexagons on unreconstructed Ag(001), as shown in Supplementary Fig. 1. Through an extensive structural search, we discovered a more stable structure, as shown in Fig. 2c. Our structure model is composed of Si dimers, analogous to that proposed by Leandri et al. However, the dimers are not tilted, and the topmost Ag layer is seriously reconstructed. Each unit cell contains two Si dimers, with one of them sinking because of the missing of two Ag atoms. The migration of Ag atoms in the topmost layer has also been found in the Si/Ag(111) and Si/Ag(110) systems[32–34], which may be a result of interaction between Si and Ag. Figure 2d shows a simulated STM image, which agrees well with previous experimental results[30]. The calculated binding energy of each Si atom (∼2.042 eV) is much larger than that of the hexagonal structure in Supplementary Fig. 1 (∼1.039 eV), indicating that our structure model is more stable. The validity of our structure model is also supported by the agreement between the calculated band structure and ARPES measurement results, as discussed in the following.

### ARPES measurements

ARPES measurements were carried out to study the electronic structure of the Si/Ag(001)-3 × 3 surface. Constant energy contours (CECs) of Si/Ag(001)-3 × 3 and pristine Ag(001) are displayed in Fig. 3a–d and Supplementary Figs. 2 and 3. All the observable bands from the Fermi surface to $E_B$ -0.5 eV are derived from the Ag(001) substrate. When the binding energy increases to 0.6 eV, two dot-like features emerge at the M point of Si, as indicated by the black arrows in Fig. 3a. With increasing binding energies, each dot becomes a closed pocket (Fig. 3b), indicating a hole-like band centered at the M point. Further increase of the binding energy leads to the touch and merge of neighboring pockets, which is a signature of a saddle point or van Hove singularity at the X point of Si.

ARPES intensity maps along Cuts 1-4 (indicated by the red lines in Fig. 3a) are displayed in Fig. 3e, g, i, j, respectively. Because Cuts 1-3 go

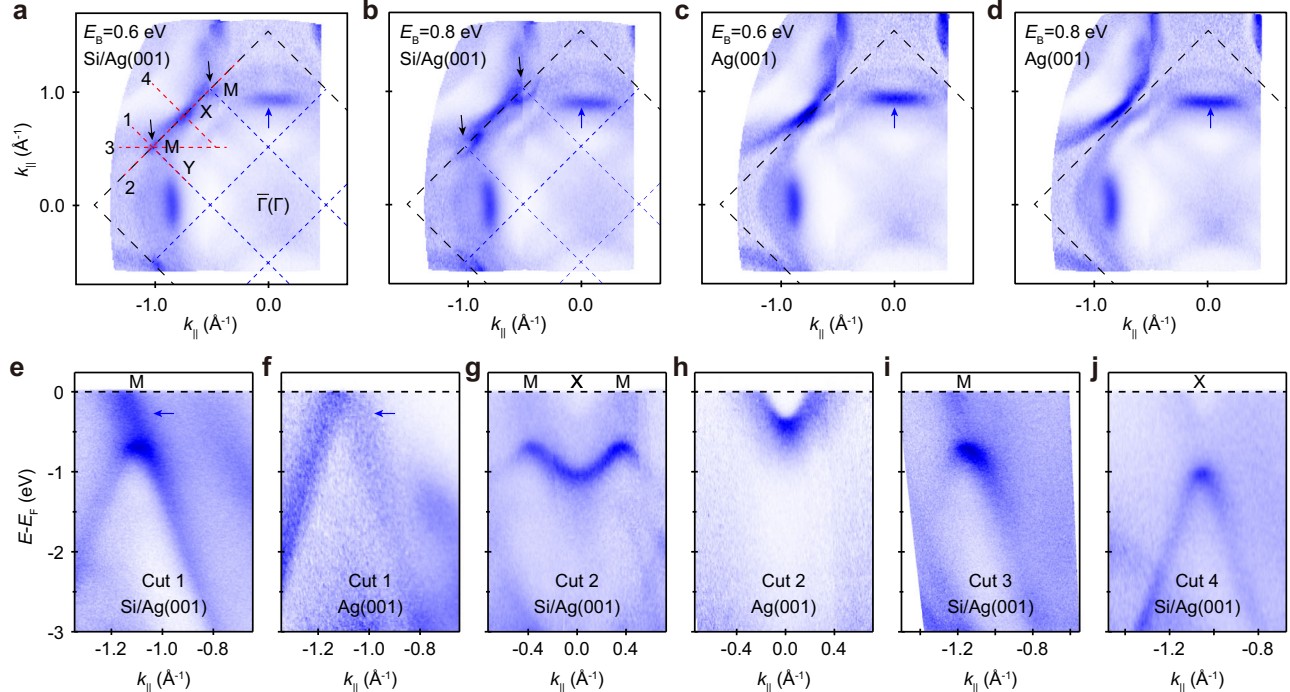

**Fig. 3 | ARPES measurement of Si/Ag(001). a, b** ARPES intensity plots of Si/Ag(001) at binding energies of 0.6 and 0.8 eV, respectively. The red lines in **a** indicate the momentum cuts where (**e**–**j**) are taken. **c, d** The same as **a** and **b** but for pristine Ag(001). **e**–**j** ARPES intensity plots along Cuts 1-4 indicated in **a**. The black and blue dashed squares in **a**–**d** indicate the BZs of Ag(001) and Si, respectively. The black arrows in panels **a, b** indicate the appearance of gapped Dirac cones. The blue arrows in **a**–**f** indicate the bulk bands of Ag(001).

through the M point, a hole-like band is observed with the band top at -0.6 eV, as shown in Fig. 3e, g, i. Along Cut 2, we observed an "M"-shaped band with a local minimum at the X point (-1.0 eV). Along the perpendicular direction, i.e., Cut 4, we observed a hole-like band. The binding energy and momentum of the band top coincide with those of the local minimum along Cut 2, which confirms the existence of a saddle point at the X point. Therefore, the electronic structure of the Si lattice is manifested by a hole-like band at the M point and a saddle point at the X point.

Based on our ARPES measurement results, we can conclude that the hole-like bands at the M point originate from the surface Si layer instead of the substrate based on the following reasons. First, these bands do not exist in pristine Ag(001) given any momentum shift, which excludes the folding effects of bulk bands. Second, because of the surface sensitivity and finite $k_z$ resolution of our ARPES measurements, the bulk bands of Ag are much blurrier, as shown in Fig. 3f, h, and are unlikely to become sharper after the growth of Si. Third, these bands behave as closed pockets at specific binding energies (Fig. 3b) and cannot be obtained by a rigid shift of the bulk bands of Ag(001). In addition, an energy shift of the bulk bands can be easily excluded because our ARPES results indicate a negligible energy shift after the growth of Si, as indicated by the blue arrows in Fig. 3a–f. We will show later that the hole-like band at each M point originates from a gapped Dirac cone, which can be well described by the long-sought 2D SSH model.

Notably, the Si-derived bands are only observable in the second BZ of Si in our ARPES measurements. This is a common phenomenon in photoemission experiments because of the matrix element effect. In addition, the bands in the second BZ of Si are closer to the bulk bands of Ag(001), and the transition rates of electrons from Ag(001) to these electronic states are much higher during the photoemission process, resulting in the stronger spectral weight. A similar phenomenon has been observed in (3 × 3)-silicene on Ag(111)[13].

## First-principles calculations and TB analysis

To understand the electronic structures of the Si lattices, we carried out first-principles calculations including both Si and the Ag(001) substrate. Using the orbital-selective band unfolding technique, we unfolded the effective band structures of Si/Ag(001) to the first BZ of Ag(001). The calculated band structures are projected onto the top two layers, i.e., the Si and topmost Ag layer, to compare with the ARPES data. The calculated band structures along the $\bar{M} - \bar{X} - \bar{M}$ direction of Ag(001) are shown in Fig. 4a. There is an "M" shaped band, as indicated by the blue dotted line. Each band top is located at 1/3 of $\bar{M} - \bar{X}$, corresponding to the M point of Si, which agrees well with our ARPES measurement results. The bands near the $\bar{X}$ point of Ag(001) have higher spectral weight, in agreement with our ARPES measurement results. In addition, neither pristine nor 3 × 3 reconstructed Ag(001) can reproduce the experimental results, as shown in Supplementary Fig. 4. Calculated band structures along Cuts 1-4 in Fig. 3 are displayed in Fig. 4b–e, and all are consistent with our ARPES measurement results. Our orbital analysis shows that the hole-like band is mainly contributed by the Si layer, instead of the Ag substrate, as shown in Supplementary Fig. 5a, b. The agreement between the calculated and experimental band structures further confirmed our structure model. The calculated binding energy of the band top (-0.9 eV) is slightly higher than the experimental value (-0.6 eV), which might originate from the different chemical potentials in real materials.

Having confirmed the atomic and electronic structures of the Si lattice, we move on to study the origin of the hole-like band at the M point by TB analysis. Our orbital analysis revealed that this band is dominated by $p_x$ and $p_z$ orbitals of Si, as shown in Supplementary Fig. 5. Therefore, we only consider the $p_x$ and $p_z$ orbitals of Si in the TB model. The Ag substrate interacts with the Si lattice and modulates the hopping magnitudes and on-site energies of Si atoms. The TB model takes

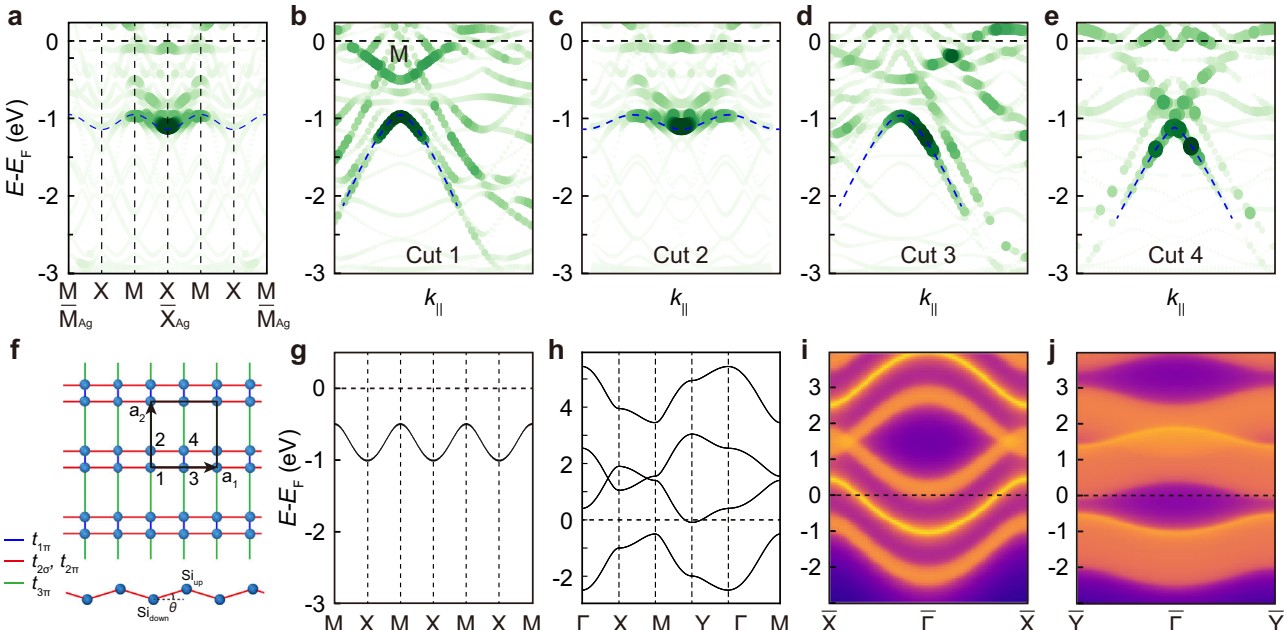

**Fig. 4 | Theoretical calculation results of Si/Ag(001). a** DFT calculated band structure of Si/Ag(001) along the $\bar{M} - \bar{X} - \bar{M}$ direction of Ag(001). **b–e** DFT calculated band structures for comparison with ARPES spectra along Cuts 1-4 in Fig. 3. The calculated band structures are projected onto the Si and topmost Ag layer. Blue dashed lines are guides to the eye for comparison with ARPES measurement results. The marker size and color represent the spectral weight that comes from the projection of the supercell wavefunction to the BZ of the primitive cell. **f** Schematic drawing of the TB model. There are four Si atoms in each unit cell, as indicated by the numbers 1-4. **g, h** Calculated band structure based on the TB model. Parameters: $t_{1\pi} = 1.2$ eV, $t_{2\sigma} = t_{2\pi} = 1.15$ eV, $t_{3\pi} = 0.25$ eV, and $U_1 = 2.5$ eV, $U_2 = 0.45$ eV. **i** Edge spectrum of the nontrivial edge which is obtained by cutting the system through the centers of the blue bonds (the Wannier center) in the $y$ direction. **j** Edge spectrum of the trivial edge by cutting the bonds in the $x$ direction.

the form:

$$
\begin{aligned}
H = - \sum_{\vec{r},(i,j),(m,n)} \Big[ & t_{1\pi} \left( a^{\dagger}_{ip_z,\vec{r}_i} a_{jp_z,\vec{r}_j} + a^{\dagger}_{ip_x,\vec{r}_i} a_{jp_x,\vec{r}_j} \right) \\
& + t_{3\pi} \left( a^{\dagger}_{ip_z,\vec{r}_i} a_{jp_z,\vec{r}_j-\vec{a}_2} + a^{\dagger}_{ip_x,\vec{r}_i} a_{jp_x,\vec{r}_j-\vec{a}_2} \right) \\
& + (\sin^2\theta t_{2\sigma} + \cos^2\theta t_{2\pi}) a^{\dagger}_{mp_z,\vec{r}_m} a_{np_z,\vec{r}_n} \\
& + (\cos^2\theta t_{2\sigma} + \sin^2\theta t_{2\pi}) a^{\dagger}_{mp_x,\vec{r}_m} a_{np_x,\vec{r}_n} \\
& + (\sin\theta\cos\theta)(-t_{2\sigma} + t_{2\pi})(a^{\dagger}_{mp_z,\vec{r}_m} a_{np_x,\vec{r}_n} + a^{\dagger}_{mp_x,\vec{r}_m} a_{np_z,\vec{r}_n}) \\
& + (\sin^2\theta t_{2\sigma} + \cos^2\theta t_{2\pi}) a^{\dagger}_{mp_z,\vec{r}_m} a_{np_z,\vec{r}_n-\vec{a}_1} \\
& + (\cos^2\theta t_{2\sigma} + \sin^2\theta t_{2\pi}) a^{\dagger}_{mp_x,\vec{r}_m} a_{np_x,\vec{r}_n-\vec{a}_1} \\
& + \sin\theta\cos\theta(-t_{2\sigma} + t_{2\pi})(a^{\dagger}_{mp_z,\vec{r}_m} a_{np_x,\vec{r}_n-\vec{a}_1} + a^{\dagger}_{mp_x,\vec{r}_m} a_{np_z,\vec{r}_n-\vec{a}_1}) \Big] \\
& + H.c. + U_1 \sum_k \left( a^{\dagger}_{kp_z,\vec{r}_k} a_{kp_z,\vec{r}_k} + a^{\dagger}_{kp_x,\vec{r}_k} a_{kp_x,\vec{r}_k} \right) \\
& + U_2 \sum_l \left( a^{\dagger}_{lp_z,\vec{r}_l} a_{lp_z,\vec{r}_l} + a^{\dagger}_{lp_x,\vec{r}_l} a_{lp_x,\vec{r}_l} \right).
\end{aligned}
\tag{1}
$$

$$ (i,j) = (1,2),(3,4); (m,n) = (1,3),(2,4); k = 1,2; l = 3,4. $$

where $a^{\dagger}_{ip_z,\vec{r}_i}$ ($a^{\dagger}_{ip_x,\vec{r}_i}$) and $a_{ip_z,\vec{r}_i}$ ($a_{ip_x,\vec{r}_i}$) are the creation and annihilation operators of the $3p_z$ orbital ($3p_x$) orbital of the $i^{th}$ Si atom located at $\vec{r}_i$, $t_{2\sigma}$ and $t_{1\pi}$, $t_{2\pi}$, $t_{3\pi}$ the $\sigma$- and $\pi$-type hopping integrals between different Si atoms, as indicated in Fig. 4f. The $\sigma(\pi)$-type hopping describes the hopping on neighboring sites with the p orbitals along (perpendicular to) the bond direction. $U_1$ ($U_2$) is the on-site energy imposed by the Ag(001) substrate on the $Si_{down}$ ($Si_{up}$) $3p_x$ and $3p_z$ orbitals. $\theta$ is the angle between the $Si_{down}$-$Si_{up}$ bond and the $x$ direction, as indicated in Fig. 4f. Interestingly, we find that $\theta$ will not be involved in the Hamiltonian, and total hopping integrals between $3p_z$ and $3p_x$ orbitals will be canceled out when $t_{2\sigma} = t_{2\pi}$. As a result, the $3p_x$ and $3p_z$ orbitals can be treated

separately and the $3p_x$ and $3p_z$ bands are the 2D SSH model, respectively. Moreover, the $3p_x$ and $3p_z$ bands become degenerate with equal on-site energies. On the other hand, the structure model in Fig. 4f provides constraints on the TB parameters. That is, hopping integrals alternate in the y direction, and on-site energies stagger in the x direction. By fitting to the experimental results, we obtain $t_{1\pi} = 1.2$ eV, $t_{2\sigma} = t_{2\pi} = 1.15$ eV, $t_{3\pi} = 0.25$ eV, $U_1 = 2.5$ eV, and $U_2 = 0.45$ eV. These parameters give rise to a gapped Dirac cone at each M point, as shown in Fig. 4g, h, which agree well with our first-principles calculation and experimental results.

To study the topology of the gapped Dirac cone, we calculate the Wannier center of the TB model, which is also the wave polarization given by the following expression:

$$ \vec{P} = \frac{1}{2\pi} \int dk_x dk_y Tr\left[\vec{A}\left(k_x, k_y\right)\right] \tag{2} $$

where $\vec{A} = \langle \Psi | i\partial_k | \Psi \rangle$ is the Berry connection integrated over the first BZ. Here, we only consider one of the $3p_x$ and $3p_z$ bands, since the $3p_x$ and $3p_z$ bands are degenerate. We obtain $\vec{P} = (0, 0.12625)$ for the gaps, indicating that the polarization is along the $y$ direction and the Wannier center is located at the center of the blue bond. To show the relationship between the polarization and edge topology, we calculate the edge spectrum of the edge in the $y$ and $x$ directions, respectively. The edge in the $y$ direction is obtained by cutting the system through the center of the blue bonds, i.e., the Wannier centers. Interestingly, nontrivial edge states emerge in the bandgap, as shown in Fig. 4i, indicating the nontrivial topology of this edge. For the edge in the $x$ direction, there are no edge states in the bulk bandgap, as shown in Fig. 4j. Therefore, the anisotropic polarization gives rise to anisotropic edge topology. The existence of topological edge states is also confirmed by first-principles calculations, as shown in Supplementary Fig. 6.

Consider a 1D SSH model. A gapless Dirac cone emerges when both the hopping integrals and on-site energies are uniform. A 2D SSH model can be viewed as two 1D SSH models in the $x$ and $y$ directions. When each 1D SSH model has a gapless Dirac cone with the Dirac points at the same binding energy, a gapless Dirac cone will emerge in the 2D SSH model. A special case is that the hopping integrals and on-site energies along the $x$ direction are the same as those along the $y$ direction, respectively, which is the model proposed in Fig. 1a. In that case, a gapless Dirac cone with isotropic Fermi velocity will emerge, as shown in Fig. 1b. When the hopping integrals are different along the $x$ and $y$ directions, the Dirac cone will become anisotropic (Supplementary Fig. 7). In the Si/Ag(001) system, the anisotropic Dirac cone is gapped out by the alternate hopping integrals in the $y$ direction and staggered on-site energies in the $x$ direction.

Finally, we discuss the role of the Ag substrate in the realization of the 2D SSH model. The Ag substrate has two major effects on the Si/Ag(001) system. First, it stabilizes the rectangular Si lattice because freestanding Si lattice is thermodynamically unstable. Second, from the viewpoint of the TB model, the effect of the Ag substrate is simplified as the on-site energy. Based on the chemical environments of Si atoms, the on-site energies stagger in one direction and keep constant in the other, giving rise to the gapped Dirac cones. If the Ag substrate is neglected, the freestanding rectangular Si lattice, although unstable, can also realize a 2D SSH model, but the detailed parameters will differ. Therefore, the Ag substrate is not a necessary condition for the realization of the 2D SSH model but an indispensable condition for our Si/Ag(001) system.

To summarize, we demonstrated the realization of the 2D SSH model based on a rectangular Si lattice grown on Ag(001). Combined ARPES measurements, first-principles calculations, and TB analysis confirmed the existence of gapped Dirac cones with an anisotropic polarization. In addition, our theoretical calculations reveal that topological edge states exist along specific peripheries. These results show that Dirac cones can exist in non-hexagonal 2D systems, which could stimulate further experimental and theoretical efforts to explore 2D topological states based on the SSH model, including topological semimetals, (high-order) topological insulators, topological crystalline insulators, and topological superconductors.

## Methods

### Experiments
The samples were prepared by evaporating Si from a Si wafer onto a Ag(001) substrate. Before growth, the Ag(001) surfaces were cleaned by repeated Ar ion sputtering and annealing cycles. The substrate temperature was kept at ~500 K during growth. The coverage of Si is approximately one monolayer, as confirmed by combined LEED and ARPES measurements. ARPES experiments were performed at 40 K with a SPECS PHOIBUS 150 electron energy analyzer and a helium discharge lamp (He Iα light). The base pressure during ARPES measurements was ~$1 \times 10^{-8}$ Pa.

### First-principles calculations
First-principles calculations based on density functional theory (DFT) were performed with the Vienna ab initio simulation package (VASP)[35,36]. The projector-augmented wave pseudopotential[37] and Perdew-Burke-Ernzerhof exchange-correlation functional[38] were used. The energy cutoff of the plane-wave basis was set as 250 eV, and the vacuum space was larger than 15 Å. The first BZ was sampled according to the Monkhorst-Pack scheme. We used a $k$ mesh of $4 \times 4 \times 1$ for structural optimization and $8 \times 8 \times 1$ for the self-consistent calculations. The positions of the atoms were optimized until the convergence of the force on each atom was <0.01 eV/Å. The convergence condition of electronic self-consistent loop was $10^{-5}$ eV.

## Data availability
The data that support the findings of this study are available within the article and Supplementary Information. Extra data are available from the corresponding authors upon reasonable request.

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

## Acknowledgements

This work was supported by the Ministry of Science and Technology of China (grants no. 2018YFE0202700 and no. 2021YFA1400502), the National Natural Science Foundation of China (Grants no. 11974391, no. 11825405, no. 1192780039, and no. U2032204), the International Partnership Program of Chinese Academy of Sciences (grant no. 112111KYSB20200012), and the Strategic Priority Research Program of Chinese Academy of Sciences (grants no. XDB33030100, and no. XDB30000000).

## Author contributions

B.F. conceived the research. D.G., S.Y., Z.S., and B.F. performed the experiments; D.G., P.C., L.C., K.W., and B.F. analyzed the experimental data; H.Z. and S.M. performed theoretical calculations; D.G., H.Z., and B.F. wrote the manuscript with comments from all authors.

## Competing interests

The authors declare no competing interests.

## Additional information

**Correspondence and requests** for materials should be addressed to Sheng Meng, Kehui Wu or Baojie Feng.

