## [Peer Review File · Nature Communications]

Observation of Gapped Dirac Cones in a Two-Dimensional Su-Schrieffer-Heeger LatticeREVIEWER COMMENTS

Reviewer #1 (Remarks to the Author):

Geng et al have reported on the electronic structure of Si on Ag(001). The system is synthesized epitaxially and measured using angle-resolved photoemission spectroscopy (ARPES). It is argued that this system is a realization of a theoretical construct known as the Su-Schrieffer-Heeger (SSH) model, which is a 2D topological system. This is substantiated by comparison to first-principle calculations and tight-binding calculations.

This is a nicely written paper, which walks the reader through the background of the SSH model and the intricacies of the Si/Ag(001) system. In principle it has the necessary novelty for Nature Communications, as the experimental realization of a model system is of fundamental significance. However, I have some concerns about the interpretations which should be clarified before I can recommend publication. In particular, the experiment itself is not decisive, since the topological signatures (edge states) are not observed. Therefore, the mapping onto the SSH model is based on the 2D band dispersions, which require further clarification. I elaborate below.

1. From the ARPES measurements (Fig 3) the authors conclude that the measured bands are associated with the Si layer. However, when comparing Si/Ag(001) to pristine Ag(001) [e.g. panels (e)/(f) and (g)/(h)] the bands look qualitatively the same, with the only difference being a rigid shift of the band structure. Is it possible that only the Ag bands are relevant, with a rigid shift due to charge transfer from the Si?
2. Following up on this point: qualitatively, the strongest effect of Si deposition is to change the symmetry due to 3x3 construction. This is quite clear in the LEED data (Fig 2a). This should have a dramatic effect in the ARPES spectra by reducing the Brillouin zone size, with associated folding. However, the measured symmetries appear fully consistent with the Brillouin zone of pristine Ag(001). This again makes me skeptical about any role of the Si in the ARPES spectra.
3. The authors seem to argue that the relevant states are primarily Si-derived. But the Ag states may also be folded by the superstructure. Therefore, the calculated band structure (Fig 4a) is quite complex and it is difficult to disentangle these different ingredients. It would be better if the authors also computed pristine Ag(001) and reconstructed Ag(001) (without Si) to clearly separate these effects. On a related note: would 3x3 folded Ag(001) also be a realization of the SSH model, or are the Si states necessary?
4. I suggest the authors show a more direct, side-by-side comparison between the first-principles calculations (or ARPES) with the tight-binding model. The comparisons in Fig 4 are very difficult to follow since different momentum trajectories are plotted. It is also not always clear whether the high-symmetry labels refer to the pristine Brillouin zone or the 3x3 Brillouin zone.
5. Do the edge states appear in the first-principles calculations? If they only appear in the tight-binding model, the argumentation is less convincing, since the tight-binding model is SSH-like by construction.
6. The surface of Ag should host Tamm surface states. Do these contribute to the electronic structure?
7. In general, the plotting of the first-principles calculations is unclear. This is partially due to the image resolution, but perhaps also due to the plotting style. Is it possible to plot in a way that makes the band dispersions more visible? Also, what do the marker sizes and colors represent?

Reviewer #2 (Remarks to the Author):

Geng and coworkers have used a combination of experimental (ARPES measurements) and theoretical methods (first-principles and tight-binding calculations) to prove the existence of gapped Dirac cones at the Brillouin zone corners of Si/Ag(001). I have no objection on the long-range order of their system (LEED) and of their ARPES results, except for some minor details that will be mentioned later.

My main concern is that all proof for the existence of Dirac cones comes from the tight-binding model that yields -as each tight-binding model- periodic bands with relatively large gaps between them. In order to be convinced that these calculated bands could indeed correspond to Dirac cones, I would suggest:

- (i) that the authors give us the tight binding parameters for which the Dirac cones would be gapless and the corresponding band diagrams
- (ii) to explain how would the parameters of (i) "move" the Si adatoms in a way that would remain consistent with the structure proposed in Fig. 1a

I also have some minor comments:

(1) The authors wrote that the simulated STM image agrees with previous results. However, there is no reference to those previous results.

(2) The ARPES E-k maps are very clear. Nevertheless, the constant energy contours are less clear. I would suggest that the authors remove -or at least make fainter- the red lines in Fig. 3a that mask the experimental data. Moreover, I think that the author needs more successive contours from 0.6 eV to 1.1 eV that follow the evolution of the contours around M from dot-like to pockets that finally merge with each other. Finally, I suggest to make zoom panels around the M points.

(3) I suggest that the authors include the label of the corresponding high-symmetry point in the band diagrams shown in Fig. 3. The scaling of the horizontal axes is sometimes confusing as $k=0.0$ refers sometimes to the Gamma and sometimes to the X point.

(4) Looking at the first-principles calculations of Fig. 4a, I would expect that more features would be experimentally seen between the M_{Si} and the X point. Do the authors have any idea why the feature with the strongest intensity (according to calculations) is not observed by means of ARPES? I refer to the weakly dispersing feature at around 1eV.

Reviewer #3 (Remarks to the Author):

In this manuscript, the authors discuss the electronic properties of rectangular Si lattice on Ag(001) by using ARPES measurement and DFT calculations. The main claim of the manuscript is that the rectangular Si lattice on Ag(001) can be modeled as the anisotropic 2D Su-Schrieffer-Heeger (SSH) model, which might be the first realization of 2D SSH square lattice in Materials Science. The claim relies on the energy band gap observation by using ARPES measurement.

The 2D SSH model is one of the recent hot topics in topological materials science. Since it shows the topological phase transition based on Zak phase. In this point, the manuscript treats the timely topic.

However, I think the analysis is relatively weak to scientifically support the evidence of realization of 2D SSH lattice to publish. The problem is following:

1. DFT part is very poor. It does not provide enough information to judge. The figure of DFT calculation is very unclear. It seems to be not unfolded, which means that it is hard to compare

with ARPES measurement.

2. The authors should discuss more carefully the effect of Ag substrate. At present, almost no data are presented in this aspect.

3. It is very unclear how the tight-binding model is derived from DFT result.

In conclusion, I cannot judge that the manuscript contains enough scientific materials. Thus, I cannot recommend the manuscript for publication.

Response to Reviewer #1

C1: *Geng et al have reported on the electronic structure of Si on Ag(001). The system is synthesized epitaxially and measured using angle-resolved photoemission spectroscopy (ARPES). It is argued that this system is a realization of a theoretical construct known as the Su-Schrieffer-Heeger (SSH) model, which is a 2D topological system. This is substantiated by comparison to first-principle calculations and tight-binding calculations.*

Reply:

We thank the reviewer for the in-depth review of our paper. The reviewer accurately summarized the main messages of our work.

C2: *This is a nicely written paper, which walks the reader through the background of the SSH model and the intricacies of the Si/Ag(001) system. In principle it has the necessary novelty for Nature Communications, as the experimental realization of a model system is of fundamental significance. However, I have some concerns about the interpretations which should be clarified before I can recommend publication. In particular, the experiment itself is not decisive, since the topological signatures (edge states) are not observed. Therefore, the mapping onto the SSH model is based on the 2D band dispersions, which require further clarification. I elaborate below.*

Reply:

We thank reviewer for the high evaluation of our work. We agree that “*the mapping onto the SSH model is based on the 2D band dispersions, which require further clarification*”. In the following, we will address all the comments one by one.

C3. *From the ARPES measurements (Fig 3) the authors conclude that the measured bands are associated with the Si layer. However, when comparing Si/Ag(001) to pristine Ag(001) [e.g. panels (e)/(f) and (g)/(h)] the bands look qualitatively the same, with the only difference being a rigid shift of the band structure. Is it possible that only the Ag bands are relevant, with a rigid shift due to charge transfer from the Si?*

Reply:

We thank the reviewer for the comment. The reviewer doubt that the Dirac-like bands might originate from the bulk bands of Ag, with a charge transfer induced energy shift. Here, we can exclude the bulk bands of Ag based on the following arguments:

- 1) Experimental identification of the Dirac-like bands as the Si-derived bands comes not only from the E-k dispersion in Fig. 3(e)-(j), but also from the constant energy contours (CECs). The Dirac-like bands behave as a closed pocket at certain binding energies, as shown in Fig. R1(b) [reproduced from Fig. 3(b) of the main text]. Similar pocket-like feature cannot be obtained by a rigid energy shift of the Ag bands.
- 2) The bulk bands of Ag(111) are still visible after the growth of Si, as indicated by the blue arrows in Fig. R1(a)-(f). When we compare (a) and (c) [or (b) and (d)],

we found that these bands have no apparent shift after the growth of Si.

- 3) As far as we know, it is difficult to shift the bulk bands by surface charge doping, especially for good conductors such as Ag. Surface doping typically has negligible effects compared to the large amounts of carriers in bulk materials, which can explain the absence of detectable energy shift of the Ag bulk bands. In addition, Si has a higher electron affinity than Ag, and therefore electrons will transfer from Ag to Si, leading to an upwards band bending of the Ag bands. However, panels (e)/(f) and (g)/(h) seem to suggest an opposite direction, although an opposite energy shift cannot explain the Dirac-like bands either.
- 4) The reviewer might notice the similarity of panels (e)/(f) and (g)/(h) with a rigid energy shift. First, we discuss (e) and (f). Apparently, the bulk bands of Ag do not move after the growth of Si, as indicated by the blue arrows. The variation of the spectral intensity of the Ag bulk bands originates from the photoemission matrix element effects and only E - k dispersions provide useful information here. Second, we discuss (g) and (h). To directly compare these bands, we superimposed the two bands together, with a rigid energy shift of the (h) towards higher binding energies, as shown in Fig. R2. There are three apparent differences: (1) the slopes of the V-shaped bands at the center; (2) the bandwidths; (3) the absence of downward dispersing bands beyond the M point in pristine Ag(001). Therefore, panels (e)/(f) and (g)/(h) cannot be explained by a rigid energy shift either.

Based on the above reasons, we can unambiguously exclude Ag bands as the origin of the Dirac-like bands.

Following the reviewer's comment, we added a sentence in the last paragraph of Section 2.2 of the main text: *“Third, these bands behave as closed pockets at specific binding energies [see Fig. 3(b)] and cannot be obtained by a rigid shift of the bulk bands of Ag(001). In addition, an energy shift of the bulk bands can be easily excluded because our ARPES results indicate a negligible energy shift after the growth of Si, as indicated by the blue arrows in Fig. 3(a)-(f).”* Figure 3 has been replaced with Fig. R1.

Fig. R1: The same as Fig. 3 of the main text but with additional blue arrows in (a)-(f) for guides to the eye. The blue arrows indicate bulk bands of Ag(001).

Fig. R2: Superimposition of the Fig. R1(g) and (h) with an energy shift of (h) towards higher binding energies. The local minimums at the center match each other after the energy shift. Apparently, the Dirac-like bands in (g) cannot be explained by a rigid energy shift of the bulk bands of the substrate.

C4. *Following up on this point: qualitatively, the strongest effect of Si deposition is to change the symmetry due to 3x3 construction. This is quite clear in the LEED data (Fig 2a). This should have a dramatic effect in the ARPES spectra by reducing the Brillouin zone size, with associated folding. However, the measured symmetries appear fully consistent with the Brillouin zone of pristine Ag(001). This again makes me skeptical about any role of the Si in the ARPES spectra.*

Reply:

We thank the reviewer for the valuable comment. We agree with the reviewer that “the measured symmetries appear fully consistent with the Brillouin zone of pristine Ag(001)”. However, this does not contradict the symmetry of the Si lattice since each Dirac cone is centered at an M point of Si. The main concern of the reviewer might be the weak spectral weight in the first BZ of Si. In our opinion, there are several reasons:

- 1) Our DFT calculations including the substrate agree well with experimental results. As shown in Fig. R3 (Reproduced from Fig. 4 of the main text), the Dirac cones are visible in the proximity of the \bar{X} point of Ag(001). However, the spectral weight near the \bar{M} point of Ag(001) is quite weak (almost invisible), although the \bar{M} point of Ag(001) is equivalent to the M point of Si.
- 2) Stronger photoemission intensity in the higher-order BZ is very common because of the matrix element effect. The bands in the first BZ might be visible with different experimental setups, such as photon energies and incident angles.
- 3) Since the Si overlayer was grounded via the Ag(001) substrate, the emitted photoelectrons were compensated by the itinerant electrons from the substrate during the photoemission process. For Ag(001), the bulk *sp* bands serve as the electron source within ~3 eV of the Fermi level. The transition rates of electrons to nearby electronic states are higher because of the smaller change of k_{\parallel} , which might result in a larger photoemission cross-section of these bands. Therefore, the

Si bands that are close to the bulk bands of Ag have stronger spectral weights. Similar effects have been observed in other surface systems, such as the 3×3 silicene/Ag(111) [PRL 122, 196801 (2019)].

In conclusion, despite of the absence of spectral weight at some M points of Si, we can still conclude that these bands are from Si layer, instead of the Ag(001) substrate.

To clarify this issue, we made the following changes in the main text:

- 1) We deleted the sentence “*Notably, only bands at the second BZ are observable, which might originate from the matrix element effects in photoemission experiments.*” in second to the last paragraph of Section 2.2.
- 2) We added a new paragraph in the end of Section 2.2: “*Notably, the Si-derived bands are only observable in the second BZ of Si in our ARPES measurements. This is a common phenomenon in photoemission experiments because of the matrix element effect. In addition, the bands in the second BZ of Si are closer to the bulk bands of Ag(001), and the transition rates of electrons from Ag(001) to these electronic states are much higher during the photoemission process, resulting in the stronger spectral weight. A similar phenomenon has been observed in (3×3) -silicene on Ag(111).*”
- 3) We added a sentence “*The bands near the \bar{X} point of Ag(001) have higher spectral weight, in agreement with our ARPES measurement results.*” in the first paragraph of Section 2.3.

Fig. R3: The same with Fig. 4(a) of the main text. Notably, the Dirac-like bands near the \bar{X} point of Ag(001) have stronger spectral weight.

C5. *The authors seem to argue that the relevant states are primarily Si-derived. But the Ag states may also be folded by the superstructure. Therefore, the calculated band structure (Fig 4a) is quite complex and it is difficult to disentangle these different ingredients. It would be better if the authors also computed pristine Ag(001) and reconstructed Ag(001) (without Si) to clearly separate these effects. On a related note: would 3×3 folded Ag(001) also be a realization of the SSH model, or are the Si states necessary?*

Reply:

We thank the reviewer for the suggestion. Following the reviewer’s request, we

calculated the band structure of pristine Ag(001) and the unfolded band structure of reconstructed Ag(001) without Si, as shown in Fig. R4. Obviously, no “M”-shaped bands exist, in contrast with the DFT calculation and ARPES measurement results. Therefore, neither pristine Ag(001) nor reconstructed Ag(001) can realize the 2D SSH model.

Based on the orbital projected band structures in Supplementary Fig. 5, we find that the Dirac bands are mainly contributed by the Si atoms. This further confirms that the 2D SSH model is realized by the Si atoms.

Following the reviewer’s suggestion, we added Fig. R4 to the Supplementary Information (Supplementary Fig. 4). We also added a sentence “*In addition, neither pristine nor 3×3 reconstructed Ag(001) can reproduce the experimental results, as shown in Supplemental Fig. 4.*” in Page 6 of the main text.

Fig. R4: (a) Calculated band structures of pristine Ag(001) along the $\bar{M} - \bar{X} - \bar{M}$ direction. The momentum range were adjusted to be consistent with Cut 2 of Fig. 3 in the main text. (b) Unfolded band structures of 3×3 reconstructed Ag(001) without Si. The effective band structure is unfolded to the first BZ of Ag(001). Four layers of Ag atoms were considered in the calculations.

C6. *I suggest the authors show a more direct, side-by-side comparison between the first-principles calculations (or ARPES) with the tight-binding model. The comparisons in Fig 4 are very difficult to follow since different momentum trajectories are plotted. It is also not always clear whether the high-symmetry labels refer to the pristine Brillouin zone or the 3x3 Brillouin zone.*

Reply:

We thank the reviewer for the valuable suggestion. In the revised manuscript, we made the following changes:

- 1) We modified Fig. 4. The DFT and TB band structures along the same momentum cuts are shown in Fig. 4(a) and (c) for a side-by-side comparison. One can see a good agreement. The Fermi levels are different because our TB parameters are fitted to experimental results.
- 2) The TB band structures along all high-symmetry directions are shown in Fig. 4(d).

C7. *Do the edge states appear in the first-principles calculations? If they only appear*

in the tight-binding model, the argumentation is less convincing, since the tight-binding model is SSH-like by construction.

Reply:

We thank the reviewer for the comments. To make our arguments more convincing, we calculated the edge spectrum by both TB and DFT, and the results are shown in Fig. R5. From the TB calculations in Fig. R5(a) and (b), the nontrivial edge has two more edge states in the proximity of the Fermi level than the trivial edge. The DFT calculation results are much more complicated than the TB calculation results because more edge and bulk bands exist. However, the DFT calculated edge spectrum of the nontrivial edge has two more edge states than that of the trivial edge, as indicated by the white arrows in Fig. R5(a). The shapes of these two bands agree well with our TB calculation results. Therefore, our DFT calculations fully support the topological nature of the 2D SSH model.

In the revised manuscript, we added Fig. R5 to the Supplementary Information (Supplementary Fig. 8). We also added a sentence “*The existence of topological edge states is also confirmed by first-principles calculations, as shown in Supplementary Fig. 8.*” during the discussion of the edge states.

Fig. R5: (a,c) TB and DFT calculated edge spectrum of the nontrivial edge which is obtained by cutting the system through the centers of the blue bonds in the y direction (the Wannier center). (b,d) TB and DFT calculated edge spectrum of the trivial edge which is obtained by cutting the system through the centers of the green bonds in the y direction. White arrows in (c) indicate topological edge states. The substrate was included in the DFT calculations.

C8. *The surface of Ag should host Tamm surface states. Do these contribute to the electronic structure?*

Reply:

We thank the reviewer for the comment. In our opinion, the Tamm surface states of Ag(001) do not contribute to the Dirac-like bands in our ARPES measurements. There are three reasons. (1) The Tamm surface states of Ag(001) exist at the \bar{M} of Ag(001). These states are split from bulk d bands by the surface potential and are located at a binding energy of ~ 3.7 eV, which is much higher compared to the energy range of our experiment (within 3 eV of the Fermi level). (2) To the best of our knowledge, the Tamm states are very sensitive to surface potential distortion, *e.g.*, adsorbates. Therefore, the Tamm states will completely disappear after growth of Si. (3) Our DFT calculation results show that the Dirac-like bands are mainly contributed

by the Si atoms, instead of the Ag atoms.

In conclusion, the Tamm states do not contribute to the Dirac bands of the Si/Ag(001) system.

References:

[Surf. Sci. 122, L629-L634 (1982)]

[Phys. Rev. B 32, 4956 (1985)]

[Surf. Sci. 178, 300-310 (1986)]

[Solid State Commun. 67, 163-167 (1988)]

[Phys. Rev. B 91, 125435 (2015)]

[Phys. Rev. B 105, L241412 (2022)]

C9. *In general, the plotting of the first-principles calculations is unclear. This is partially due to the image resolution, but perhaps also due to the plotting style. Is it possible to plot in a way that makes the band dispersions more visible? Also, what do the marker sizes and colors represent?*

Reply:

We thank the reviewer for the suggestion to help improve the data presentation. We replot our first-principles calculation results in green color and with 1.5 times of weight, as shown in Fig. 4(a) of the revised manuscript.

In the following, we explain the meaning of the marker size and color. The unfolding of the band structure gives a spectral function (not $\varepsilon(k_i)$ dispersion relation):

$$A(k_i, \varepsilon) = \sum_N P(k_i; K, N) \delta(\varepsilon - \varepsilon(N, K)),$$

where $P(k_i; K, N) = \sum_n \langle \Psi_{N,K} | \varphi_{n,k_i} \rangle \langle \varphi_{n,k_i} | \Psi_{N,K} \rangle$, $\Psi_{N,K}$ and φ_{n,k_i} are the supercell (SC) and primitive cell (PC) wavefunction, respectively, N and n are the band index of SC and PC, respectively, and K and $k_i = K + G$ (G is the reciprocal vector of SC) are the momentum of SC and PC, respectively. Therefore, the marker size and color represent the weight that comes from the projection of SC wavefunction to the BZ of the PC. The marker size and color are related to enhancing the primary electronic bands. That is, the larger the size, the darker the color.

When these bands are projected to the orbitals of specific atoms, an extra weight function $W(N, K)$, *i.e.*, the projected density of states of certain orbitals, should be included in the spectral function:

$$A(k_i, \varepsilon) = \sum_N P(k_i; K, N) W(N, K) \delta(\varepsilon - \varepsilon(N, K)).$$

The marker size and color represent the product of the weight that comes from the projection of the SC wavefunction to the BZ of the PC, and the weight comes from the projection of the SC wavefunction to the orbitals of specific atoms.

In the revised manuscript, we added a sentence “*The marker size and color represent the spectral weight that comes from the projection of the supercell wavefunction to the BZ of the primitive cell.*” to the caption of Fig. 4.

Response to Reviewer #2

C1: *Geng and coworkers have used a combination of experimental (ARPES measurements) and theoretical methods (first-principles and tight-binding calculations) to prove the existence of gapped Dirac cones at the Brillouin zone corners of Si/Ag(001). I have no objection on the long-range order of their system (LEED) and of their ARPES results, except for some minor details that will be mentioned later.*

Reply:

We thank the reviewer for the in-depth review and high evaluation of our work. In the following, we will address all the comments one by one.

C2: *My main concern is that all proof for the existence of Dirac cones comes from the tight-binding model that yields -as each tight-binding model- periodic bands with relatively large gaps between them. In order to be convinced that these calculated bands could indeed correspond to Dirac cones, I would suggest:*

- (i) that the authors give us the tight binding parameters for which the Dirac cones would be gapless and the corresponding band diagrams*
- (ii) to explain how would the parameters of (i) “move” the Si adatoms in a way that would remain consistent with the structure proposed in Fig. 1a*

Reply:

We thank the reviewer for the suggestions.

- (i) To obtain a gapless Dirac cone, we set the parameters: $t_{1\pi} = t_{3\pi} = 0.25$ eV, $U_1 = U_2 = 0.45$ eV, and $t_{2\sigma} = t_{2\pi} = 1.15$ eV. With these parameters, the hopping integrals in either x and y directions are the same, and the on-site potentials of the two Si atoms in either x and y direction are also the same. These parameters will give rise to a gapless Dirac cone, as shown in Fig. R6. Since the hopping integrals in the x direction are different from those of the y direction, the Dirac cone is anisotropic, *i.e.*, the Fermi velocities are different along the x and y directions.
- (ii) Consider a 1D SSH model. A gapless Dirac cone emerges when both the hopping integrals and on-site energies are uniform. A 2D SSH model can be viewed as two 1D SSH models in the x and y directions, respectively. To obtain a gapless Dirac cone in the 2D SSH model, both 1D SSH models should have a gapless Dirac cone with the Dirac point at the same binding energy. That is to say, along each direction, the hopping integrals should be equal, as well as the on-site energies. A special case is that the hopping integrals and on-site energies along the x direction are the same as those along the y direction, respectively, which is the model proposed in Fig. 1(a). In that case, a gapless Dirac cone with isotropic Fermi velocity will emerge, as shown in Fig. 1(b) of the main text. When the hopping integrals are different along the x and y directions, the Dirac cone will become anisotropic, as shown in Fig. R6. In the Si/Ag(001) system, the alternate hopping integrals and staggered on-site energies result in an anisotropic

and gapped Dirac cone.

Following the reviewer's suggestion, we added a new paragraph in Page 8 of the main text: "Consider a 1D SSH model. A gapless Dirac cone emerges when both the hopping integrals and on-site energies are uniform. A 2D SSH model can be viewed as two 1D SSH models in the x and y directions. When each 1D SSH model has a gapless Dirac cone with the Dirac points at the same binding energy, a gapless Dirac cone will emerge in the 2D SSH model. A special case is that the hopping integrals and on-site energies along the x direction are the same as those along the y direction, respectively, which is the model proposed in Fig. 1(a). In that case, a gapless Dirac cone with isotropic Fermi velocity will emerge, as shown in Fig. 1(b). When the hopping integrals are different along the x and y directions, the Dirac cone will become anisotropic, as shown in Supplementary Fig. 6. In the Si/Ag(001) system, the alternate hopping integrals and the staggered on-site energies results in an anisotropic and gapped Dirac cone." We also added Fig. R6 to the Supplementary Information (Supplementary Fig. 6).

Fig. R6: TB calculated band structure with parameters: $t_{1\pi} = t_{3\pi} = 0.25$ eV, $t_{2\sigma} = t_{2\pi} = 1.15$ eV and $U_1 = U_2 = 0.45$ eV. A gapless and anisotropic Dirac cone appears at the M point.

C3: *I also have some minor comments:*

(1) The authors wrote that the simulated STM image agrees with previous results. However, there is no reference to those previous results.

Reply:

We thank the reviewer for pointing out our carelessness. The reference should be Ref. [30]. We have cited this paper here in the revised manuscript.

C4: *(1) The ARPES E-k maps are very clear. Nevertheless, the constant energy contours are less clear. I would suggest that the authors remove -or at least make fainter- the red lines in Fig. 3a that mask the experimental data. Moreover, I think that the author needs more successive contours from 0.6 eV to 1.1 eV that follow the evolution of the contours around M from dot-like to pockets that finally merge with each other. Finally, I suggest to make zoom panels around the M points.*

Reply:

We thank the reviewer for the suggestions to improve the data presentation. We followed the reviewer's suggestion and made the following changes:

- 1) In Fig. 3(a), we changed the solid lines to dashed lines;
- 2) The thicknesses of both black and red lines in Fig. 3(a)-(d) are reduced. We did

not remove the red lines because they provide guides for the eye for the E - k maps in Fig. 3(e)-(j).

- 3) We added a new figure in the Supplementary Information (Supplementary Fig. 3) to show the evolution of zoomed-in constant energy contours (CECs) from 0.5 to 1.2 eV. The figure is duplicated as Fig. R7. All the red lines were removed.

Fig. R7: Zoomed-in constant energy contours near the \bar{X} point of Ag(001) from $E_B=0.5$ eV to $E_B=1.2$ eV. Dot-like features emerge from 0.6 eV and become pockets at higher binding energies. Finally, neighboring pockets merge at approximately 1.0 eV. Black arrows indicate the evolution of the Dirac cones.

C5: *I suggest that the authors include the label of the corresponding high-symmetry point in the band diagrams shown in Fig. 3. The scaling of the horizontal axes is sometimes confusing as $k=0.0$ refers sometimes to the Gamma and sometimes to the X point.*

Reply:

We thank the reviewer for the suggestion. We added labels of high-symmetry points of Si in the revised Fig. 3(e,g,i,j). Figure 3(f) and (h) were not modified because pristine Ag(001) does not have these high symmetry points.

C6: *Looking at the first-principles calculations of Fig. 4a, I would expect that more features would be experimentally seen between the M_{Si} and the X point. Do the authors have any idea why the feature with the strongest intensity (according to calculations) is not observed by means of ARPES? I refer to the weakly dispersing feature at around 1 eV.*

Reply:

We thank the reviewer for the valuable comment, which helped us to correct a mistake.

After a careful check of the DFT calculation results, we found that the position of the blue dashed lines (guides for the eye) should be readjusted, as shown in Fig. R8(a) and (b). The feature with the strongest intensity corresponds to the Dirac-like bands

that are observed by ARPES. The feature with weaker intensity, as indicated by the blue arrows in Fig. R8(b), might originate from the finite thickness of our slab calculations since this feature seems parallel with the strongest feature.

To further confirm the gapped Dirac cones in DFT calculations, we present the unfolded band structures along Cuts 1-4 that were measured by ARPES (Fig. 3 of the main text), as shown in Fig. R9. One can see that the calculated bands agree well with our ARPES measurements except that the calculated Fermi level should be slightly adjusted and the bands is a little deformed. The slight inconsistency may be caused by the following reasons. (1) The finite thickness in our DFT calculations because of the limited calculation resources; (2) A surface adsorption system is rather complex and the large number of atoms in our calculations may make the error larger. However, the qualitative agreement between our DFT calculation and ARPES measurement results can still provide compelling evidence for the existence of gapped Dirac cones in this system.

In the revised manuscript, we have corrected the blue dashed lines in Fig. 4 and Supplementary Fig. 5. In addition, we modified the first and second paragraphs in Section 2.3 to explain this issue (see the revised manuscript). We sincerely thank the reviewer for the valuable comment, which helped us to avoid mistakes.

Fig. R8: (a) The same with Fig. 4(a) in the main text (previous version). (b) The same with Fig. 4(a) in the main text except that the blue dashed line is readjusted.

Fig. R9: DFT calculated band structures for comparison with ARPES spectra along Cuts 1-4 (Fig. 3 of the main text). The calculated band structures are projected onto the Si and topmost Ag layer. The red dashed lines are guides for the eye to compare with ARPES measurement results.

Response to Reviewer #3

C1: *In this manuscript, the authors discuss the electronic properties of rectangular Si lattice on Ag(001) by using ARPES measurement and DFT calculations. The main claim of the manuscript is that the rectangular Si lattice on Ag(001) can be modeled as the anisotropic 2D Su-Schrieffer-Heeger (SSH) model, which might be the first realization of 2D SSH square lattice in Materials Science. The claim relies on the energy band gap observation by using ARPES measurement.*

The 2D SSH model is one of the recent hot topics in topological materials science. Since it shows the topological phase transition based on Zak phase. In this point, the manuscript treats the timely topic.

Reply:

We thank the reviewer for the in-depth review and high evaluation of our paper. In the following, we will address all the comments one by one.

C2: *However, I think the analysis is relatively weak to scientifically support the evidence of realization of 2D SSH lattice to publish. The problem is following:*

1. *DFT part is very poor. It does not provide enough information to judge. The figure of DFT calculation is very unclear. It seems to be not unfolded, which means that it is hard to compare with ARPES measurement.*

Reply:

We thank the reviewer for the comments. We confirm that the DFT calculated band structures were unfolded. The reviewer may be confused by the poor presentation. To improve the presentation of the DFT calculation results, we made the following changes:

- (1) We replotted our DFT calculations in green color with 1.5 times of weight, as shown in the new Fig. 4(a).
- (2) In the revised Supplementary Information, we presented unfolded band structures (Supplementary Fig. 6) along Cuts 1-4 that were measured by ARPES in Fig. 3 of the main text, as shown in Fig. R10. These results can be directly compared with our ARPES results.

We believe that the DFT part in the revised manuscript has been significantly improved. The DFT calculation results are much more convincing now.

Fig. R10: DFT calculated band structures for comparison with ARPES spectra along

Cuts 1-4 (Fig. 3 of the main text). The calculated band structures are projected onto the Si and topmost Ag layer. The red dashed lines are guides for the eye to compare with ARPES measurement results.

C3. 2. *The authors should discuss more carefully the effect of Ag substrate. At present, almost no data are presented in this aspect.*

Reply:

We thank the reviewer for the comment. We agree that the effect of the Ag substrate should be discussed.

The Ag substrate has two major effects on the Si/Ag(001) system. First, it stabilizes the rectangular Si lattice because freestanding Si lattice is thermodynamically unstable. Second, from the viewpoint of the TB model, the effect of the Ag substrate is not neglected, but simplified as the on-site energy. Based on the chemical environments of Si atoms, the on-site energies stagger in one direction and keep constant in the other, giving rise to gapped Dirac cones.

If the Ag substrate is neglected, the freestanding rectangular Si lattice, although unstable, can also realize a 2D SSH model, but the detailed parameters will differ. Therefore, the Ag substrate is not a necessary condition for the realization of the 2D SSH model but an indispensable condition for our Si/Ag(001) system.

Following the reviewer's suggestion, we added a paragraph before the Summary Section: *“Finally, we discuss the role of the Ag substrate in the realization of the 2D SSH model. The Ag substrate has two major effects on the Si/Ag(001) system. First, it stabilizes the rectangular Si lattice because freestanding Si lattice is thermodynamically unstable. Second, from the viewpoint of the TB model, the effect of the Ag substrate is simplified as the on-site energy. Based on the chemical environments of Si atoms, the on-site energies stagger in one direction and keep constant in the other, giving rise to the gapped Dirac cones. If the Ag substrate is neglected, the freestanding rectangular Si lattice, although unstable, can also realize a 2D SSH model, but the detailed parameters will differ. Therefore, the Ag substrate is not a necessary condition for the realization of the 2D SSH model but an indispensable condition for our Si/Ag(001) system.”*

C4. *It is very unclear how the tight-binding model is derived from DFT result.*

Reply:

We thank the reviewer for the comment. The tight-binding model is derived from the DFT results by the following steps.

- (1) We confirm that our DFT calculations agree with ARPES results. Then we calculated the orbital projected band structures, as shown in Supplementary Fig. 5 of the Supplementary Information. After a careful comparison, we found that the Dirac bands observed by ARPES are primarily contributed by the Si p_x and p_z orbitals. Therefore, we only consider these two orbitals in our TB analysis.
- (2) Based on the atomic structure of the rectangular Si lattice [Fig. 4(b)], we found that the bond lengths of Si alternate in the y direction, resulting in alternate hopping integrals in the y direction. On the other hand, the Si atoms in the y

direction have the same chemical environment, resulting in uniform on-site energies in the y direction. In the x direction, the bond lengths of Si atoms are the same, resulting in uniform hopping integrals. On the other hand, the Si atoms have alternate chemical environments, resulting in staggered on-site energies in the x direction. With these constraints on the parameters, the TB model will produce gapped Dirac cones at the M point.

- (3) Finally, we adjust the values of the TB parameters to make the TB band structures agree with the ARPES results.

Following the reviewer's comment, we added a sentence "*On the other hand, the structure model in Fig. 4(b) provides constraints on the TB parameters. That is, hopping integrals alternate in the y direction and on-site energies stagger in the x direction.*" in Page 7 of the main text. This sentence explains the constraints on the TB parameters.

C5: *In conclusion, I cannot judge that the manuscript contains enough scientific materials. Thus, I cannot recommend the manuscript for publication.*

Reply:

We thank the reviewer for providing valuable comments and suggestions, which helped to improve our manuscript's quality. We added additional calculation results and discussions in the revised manuscript. We believe the revised version reached the standard for publication in *Nature Communications*.

Summary of Changes:

1. All changes in the main text are highlighted in "Revised Manuscript (Tracked)". A clear version is also appended.
2. Figures 3 and 4 were replaced.
3. Five new figures were added in the Supplementary Information: Fig. 3,4,6,7,8. The order of the figures was updated.

REVIEWERS' COMMENTS

Reviewer #1 (Remarks to the Author):

The authors have satisfactorily addressed most of my questions, especially regarding the role of the Si substrate and the interpretation of the ARPES data. The comparison to theory is clearer, and overall the manuscript is much improved.

My main overall assessment— that the experiment itself is not decisive— still stands. However, the authors have made a compelling argument that their experiment supports the SSH model based on their tight-binding and first-principles calculations. Due to the nature of 1D edge modes, it would be difficult (or impossible) to obtain more compelling evidence using ARPES. Whether this result is sufficient for a high-impact publication is a matter of subjective judgement. Taking these factors into consideration, my conclusion is to recommend publication in Nature Communications.

Some final optional suggestions for the authors to consider:

1. I found Supplementary Figure 6 very helpful for comparing to the ARPES data in Fig. 3 of the main text. I would suggest to include it in the main text.
2. I still found the unfolded calculations very messy and difficult to interpret. The marker color seems to randomly fluctuate at different k points. It is unclear if this is an artifact or a physical result.

Reviewer #2 (Remarks to the Author):

In the revised version of their manuscript Geng et al. have satisfactorily replied to the comments of mine and of the other reviewers. I appreciate the fact that the authors have expanded their manuscript with new Figures in the suppl. info and they have performed additional calculations after the reviewers' request. I have no further comments and I recommend publication in Nature communications.

Reviewer #3 (Remarks to the Author):

The authors have properly revised the manuscript, especially concerning the part of the density functional theory and tight-binding model. The revision is satisfactory. Now, I recommend the manuscript for publication.

Response to Reviewer #1

C1: *The authors have satisfactorily addressed most of my questions, especially regarding the role of the Si substrate and the interpretation of the ARPES data. The comparison to theory is clearer, and overall the manuscript is much improved.*

My main overall assessment— that the experiment itself is not decisive— still stands. However, the authors have made a compelling argument that their experiment supports the SSH model based on their tight-binding and first-principles calculations. Due to the nature of 1D edge modes, it would be difficult (or impossible) to obtain more compelling evidence using ARPES. Whether this result is sufficient for a high-impact publication is a matter of subjective judgement. Taking these factors into consideration, my conclusion is to recommend publication in Nature Communications.

Reply:

We are pleased that the reviewer is satisfied with our previous reply.

C2: *Some final optional suggestions for the authors to consider:*

1. I found Supplementary Figure 6 very helpful for comparing to the ARPES data in Fig. 3 of the main text. I would suggest to include it in the main text.

Reply:

We thank the reviewer for the suggestion. We followed the reviewer's suggestion and included Supplementary Fig. 6 (previous version) to Fig. 4 of the main text.

C3: *2. I still found the unfolded calculations very messy and difficult to interpret. The marker color seems to randomly fluctuate at different k points. It is unclear if this is an artifact or a physical result.*

Reply:

We thank the reviewer for the comment. After a careful check of the unfolded calculations, we found that the problem was caused the plotting style of the figures: light-colored circles obscure dark-colored circles, which makes continuous bands appear choppy. To solve this problem, we reversed the stacking order so that darker circles are on the top. We redraw all the unfolded band structures in the manuscript, including Fig. 4 and Supplementary Figs. 1,4,5, and believe the new figures are much more convincing.

Summary of Changes:

1. All changes in the main text are highlighted in "Revised Manuscript (Tracked)". A clear version is also appended.
2. Figure 4 were replaced. Four new panels (b-e, from Supplementary Fig. 6 of the previous version) were included following the reviewer's suggestion. The original Supplementary Fig. 6 were deleted.
3. The bottom panel of Fig. 2(c) was adjusted so that the Si atoms are aligned with the those in the upper panel.